

# Floating nurseries? Scyphozoan jellyfish, their food and their rich symbiotic fauna in a tropical estuary

José M. Riascos[1], Willington Aguirre[2], Charlotte Hopfe[3], Diego Morales[1], Ángela Navarrete[1] and José Tavera[1]

[1] Instituto de Ciencias del Mar y Limnología, Universidad del Valle, Cali, Colombia
[2] Bahía Málaga, Consejo Comunitario Comunidad Negra de La Plata Bahía Málaga, Buenaventura, Colombia
[3] Department of Biomaterials, Universität Bayreuth, Bayreuth, Germany

Corresponding author
José M. Riascos,
jose.m.riascos@correounivalle.edu.co

## ABSTRACT

**Background**. The anthropogenic modification of trophic pathways is seemingly prompting the increase of jellyfish populations at the expense of planktivorous fishes. However, gross generalizations are often made because the most basic aspects of trophic ecology and the diverse interactions of jellyfish with fishes remain poorly described. Here we inquire on the dynamics of food consumption of the medusoid stage of the scyphozoan jellyfish *Stomolophus meleagris* and characterize the traits and diversity of its symbiotic community.

**Methods**. *S. meleagris* and their associated fauna were sampled in surface waters between November 2015 and April 2017 in Málaga Bay, an estuarine system at the Colombian Pacific. Stomach contents of medusae were examined and changes in prey composition and abundance over time analysed using a multivariate approach. The associated fauna was identified and the relationship between the size of medusae and the size those organisms tested using least-square fitting procedures.

**Results**. The presence of *S. meleagris* medusa in surface waters was seasonal. The gut contents analysis revealed that algae, copepods and fish early life stages were the more abundant items, and PERMANOVA analysis showed that the diet differed within the seasons ($P_{(\text{perm})} = 0.001$) but not between seasons ($P_{(\text{perm})} = 0.134$). The majority of the collected medusae (50.4%) were associated with individuals of 11 symbiotic species, 95.3% of them fishes, 3.1% crustaceans and 1.6% molluscs. Therefore, this study reports 10 previously unknown associations. The bell diameter of *S. meleagris* was positively related to the body sizes of their symbionts. However, a stronger fit was observed when the size relationship between *S. meleagris* and the fish *Hemicaranx zelotes* was modelled.

**Discussion**. The occurrence of *S. meleagris* was highly seasonal, and the observed patterns of mean body size through the seasons suggested the arrival of adult medusae to the estuary from adjacent waters. The diet of *S. meleagris* in the study area showed differences with previous reports, chiefly because of the abundance of algae that are seemingly ingested but not digested. The low number of zooplanktonic items in gut contents suggest the contribution of alternative food sources not easily identifiable. The observed changes in the composition of food in the guts probably reflect seasonal changes in the availability of prey items. The regular pattern in the distribution of symbionts among medusae (a single symbiont per host) and the positive host-symbiont size relationship reflects antagonistic intraspecific and interspecific behaviour of the symbiont. This strongly suggest that medusa represent an "economically defendable

resource" that potentially increases the survival and recruitment of the symbionts to the adult population. We argue that, if this outcome of the symbiotic association can be proven, scyphozoan jellyfish can be regarded as floating nurseries.

# INTRODUCTION

The magnitude and frequency of population blooms of jellyfish (pelagic cnidarian and ctenophores) are seemingly increasing, along with strong impacts on marine ecosystems. The collapse of formerly rich fisheries has been linked to increasing jellyfish populations in several regions (*Lynam et al., 2006*; *Brodeur, Ruzicka & Steele, 2011*). However, much of the societal and even scientific perception about jellyfish and their role in ecosystems is based on speculation, limited evidence and flawed scientific practices (e.g., *Haddock, 2008*; *Richardson et al., 2009*; *Sanz-Martín et al., 2016*). Trophic interactions between jellyfish and planktivorous fishes have been characterized as a combination of mutual predation and competition for planktonic food (*Purcell & Arai, 2001*; *Lynam et al., 2006*; *Brodeur et al., 2008*; *Richardson et al., 2009*). Changes in the balance of these trophic pathways in stressed and overfished ecosystems have been hypothesized to explain massive local proliferations of jellyfish that displace planktivorous fishes and form alternate jellyfish-dominated ecosystems (*Richardson et al., 2009*). However, trophic relationships are only known for a small portion of this polyphyletic assemblage spanning more than 2000 species (*Fleming et al., 2015*). As a result, trophic models assessing food web structure and energy flow often ignore jellyfish or include them as a single functional group with the characteristics of an 'average' jellyfish whose parameterization frequently varies greatly among models (*Pauly et al., 2009*). In fact, the hypotheses proposed to explain changes in jellyfish dominated ecosystems remain untested, partially because there is a recognition that more basic research on feeding ecology is still required (*Pauly et al., 2009*; *Richardson et al., 2009*; *Naman et al., 2016*).

Regarding scyphozoan jellyfish, two emerging issues are challenging the predominant view of competitive trophic interactions between planktivorous fishes and jellyfish. First, recent methodological approaches have shown that scyphozoans use unsuspected food sources, including benthic organisms (*Pitt et al., 2008*; *Ceh et al., 2015*), microplankton and resuspended organic matter (*Javidpour et al., 2016*). These food sources have been traditionally overlooked, because most studies on feeding ecology use gut content analysis that focuses on mesozooplankton and ichthyoplankton, presumably because they are more visible and retained in the gut for longer than other type of food (*Pitt, Connolly & Meziane, 2009*). Thus, the widely held view that fishes and scyphozoan jellyfish compete for the same food source seems a gross generalization. Second, mounting evidence suggests that mutual predation (i.e., medusae predating on fish egg or larvae and fishes predating on medusae early life stages) is only one side of the story. Scyphozoans usually display

"symbiotic associations", defined as any living arrangement, including positive and negative associations, between members of two different species (see *Martin & Schwab, 2013*). Symbionts are diverse, ranging from fish to invertebrates, in a variety of relationships including parasitism, mutualism and commensalism (*Riascos et al., 2013*; *Ohtsuka et al., 2009*; *Ingram, Pitt & Barnes, 2017*). However, trophic modelling efforts traditionally focus on predation and competition, despite mounting evidence showing that alternative trophic pathways and relationships may positively affect fish populations (e.g., *Lynam & Brierley, 2007*; *Riascos et al., 2012*; *Greer et al., 2017*).

Much of the prevalent view about fish-jellyfish trophic dynamics is derived from scyphozoans of subtropical and temperate areas that support large pelagic fisheries. In estuarine systems, the scarce evidence suggests a low trophic overlap between fish and jellyfish (*Nagata et al., 2015*; *Naman et al., 2016*) and a high occurrence of symbiotic associations with juveniles of fish and invertebrates (e.g., *Rountree, 1983*; *Costa, Albieri & Araújo, 2005*; *Martinelli et al., 2008*), perhaps reflecting the abundance of early stages of coastal fauna in nursery areas. Hence, there is a need to study the trophic ecology of scyphozoan jellyfish and their multiple biological interactions to truly understand population dynamics, their position in food webs and their functional role in estuarine ecosystems.

Here we studied the canonball jellyfish *Stomolophus meleagris* (Agassiz, 1862), in the estuarine ecosystem of Malaga Bay, an area of high biodiversity in the Colombian pacific coast. This species is widely distributed in the western Atlantic (United States to Brazil) and the Pacific oceans (southern California to Ecuador; Sea of Japan to South China Sea) (*Calder, 1982*; *Griffin & Murphy, 2011*) and has been described as a specialized predator of fish eggs, copepods and mollusc larvae, with the capacity to regulate local populations of its prey (*Larson, 1991*). Therefore, the aims of this study were (i) to assess changes in the structure of the diet in the study area and (ii) analyse the traits, diversity and significance of the symbiotic associations in the estuarine ecosystem.

## MATERIALS & METHODS

### Sampling

The study was performed in Málaga Bay, a south-facing bay located in the central region of the Colombian Pacific coast (4°05′N and 77°16′W, Fig. 1). The bay is located within the Chocó-Darien region, an area with one of the highest levels of precipitation in the western hemisphere (7,000–11,000 mm; *Poveda, Jaramillo & Vallejo, 2014*), which has two wet seasons during the year: April–June and September–November. The water depth in this bay averages 13 m but reaches a maximum of 40 m. Tides are semi-diurnal, with a mean tidal range of 4.1 m. Sea surface temperature varies between 25 and 30 °C and salinity between 19 and 28 in the mouth of the bay and 1.3 and 10 close to small rivers (*Lazarus & Cantera, 2007*). Sampling of *S. meleagris* was conducted around La Plata Archipielago, at the innermost part of Málaga Bay. Samplings were allowed by the Autoridad Nacional de Licencias Ambientales (ANLA; permit number 1070_28-08-2015). The medusoid phase showed a seasonal occurrence in surface waters: December 2015 to May 2016 (hereafter season 1) and December 2016 to April 2017 (hereafter season 2). Sampling effort was
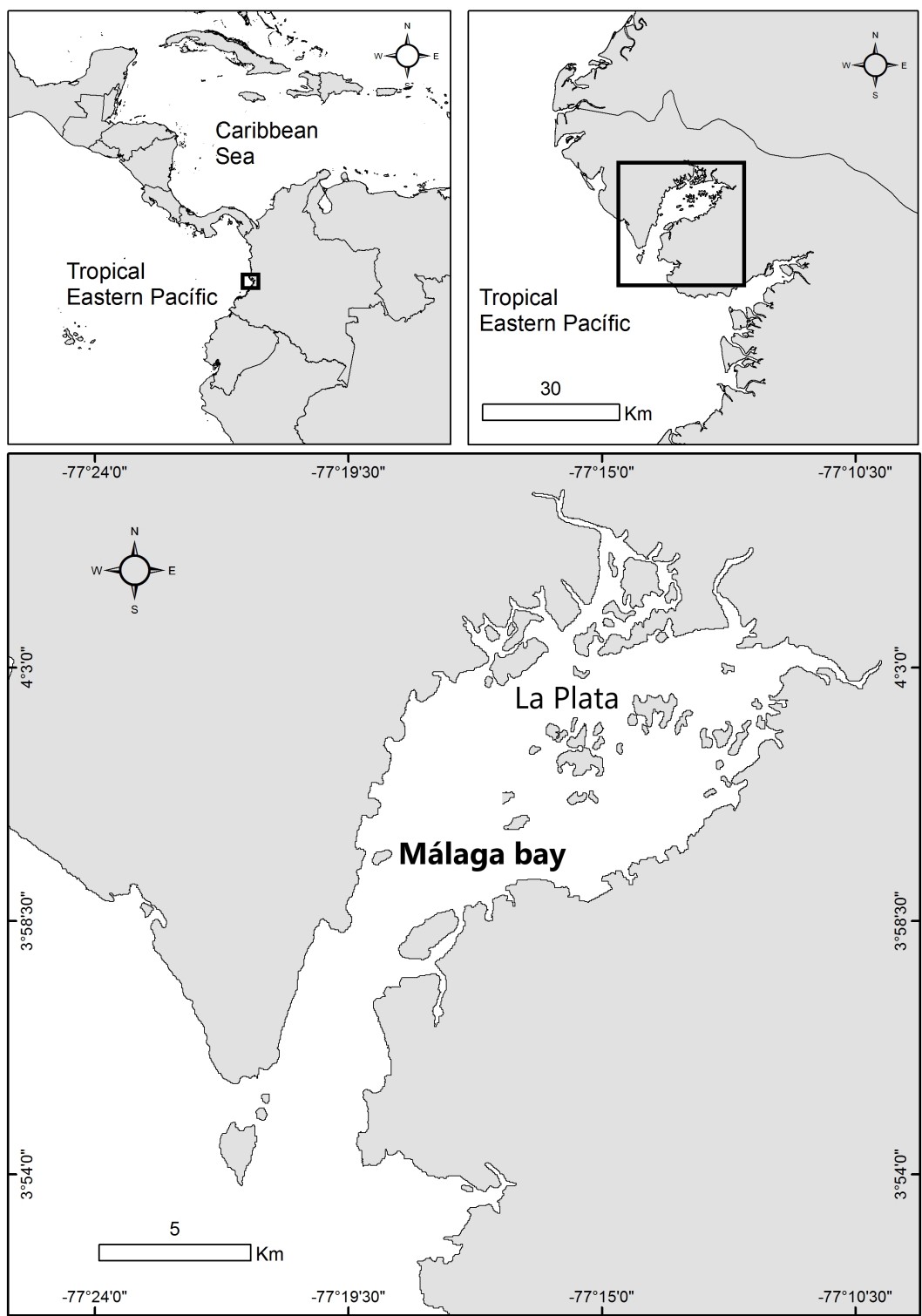

**Figure 1** **Map of the study area in La Plata Archipelago (Málaga Bay, Colombian Pacific coast).**

relatively constant throughout the seasons: when medusae were detected, three or four three-hour sighting trips were performed per month. Medusae were sampled from a small boat using dip nets, and the relative abundance was estimated as the number of caught medusae per hour. As medusas where generally associated to juvenile fish or invertebrates the sampling was limited to 15 to 20 medusae per month, to avoid disturbing populations of fish potentially under conservation. Upon collection, medusae discharged a sticky mucus that rapidly killed fish or associated invertebrates as reported by *Shanks & Graham (1988)*. Thereafter, medusae and their associated fauna were tagged and stored in individual jars with 5% borate buffered formaldehyde solution in seawater.

In the laboratory, the bell diameter of each medusa and the standard length of associated fish and invertebrates (crustaceans and molluscs) were measured using callipers. The medusa's fused oral arms and the mouth folds were excised and rinsed through a 100-$\mu$m mesh sieve to concentrate food particles. The resulting material was transferred to ethanol and sorted for the presence of prey items using a dissecting microscope. Prey items were determined to a taxonomic level suitable for making meaningful comparisons with similar studies (*Larson, 1991*; *Padilla-Serrato et al., 2013*; *Álvarez Tello, López-Martínez & Lluch-Cota, 2016*).

Associated fish were identified to species according to current keys and by comparison of their morphological features against available descriptions (*Jordan & Evermann, 1898*; *Allen & Robertson, 1994*; *Fischer et al., 1995*; *Chirichigno & Cornejo, 2001*; *Robertson & Allen, 2015*). Molluscs and crustaceans were identified by Cantera JR and JF Lazarus, respectively. Finally, all the associated fauna was stored as reference material in the scientific collection at the Marine Biology section at Universidad del Valle.

## Data analyses

A multivariate approach was used to test changes in the structure (composition and abundance) of the diet over time. For this, data were arranged in a matrix of abundance of each taxon (rows) eaten by individual medusae in each month (columns). Prey taxa with few occurrences (<0.1% of total abundance; Table 1), were excluded from further analyses. Prior to the analyses data were standardized to account for the difference in food quantity associated to distinct medusae body sizes, by dividing the abundance of each prey item by the total abundance of prey for each medusa. Moreover, data were square root transformed to slightly downweigh the contribution of abundant food items.

Non-metric multidimensional scaling (nMDS; *Clarke & Gorley, 2006*) was used to build an ordination plot of medusae per month and season calculated from a Bray–Curtis matrix of similarity in diet composition. The distance-based permutational multivariate analysis of variance (PERMANOVA, *Anderson, 2001*) was used to test for temporal differences in the structure of diet between months and seasons. The model used ''month'' as a random factor nested within the fixed factor ''season''. As the PERMANOVA approach is sensitive to differences in multivariate dispersion within groups, the PERMDISP routine was used to test for homogeneity of dispersions. Thereby, a preliminary analysis showed that multivariate dispersion was strongly dependent of sample size, with less dispersion observed at the beginning of the season, when medusae were scarce in the field and thus

Riascos et al. (2018), *PeerJ*, DOI 10.7717/peerj.5057

**Table 1** Composition and mean abundance (number of food item per medusa and standard error in brackets) of food items in gut contents of *Stomolophus melea-gris* during two seasons in Bahía Málaga, Pacific coast of Colombia.

| Food ítem | Season 1 (2015–2016) | | | | | | Season 2 (2017) | | | | Tot. | Rel. Ab (%) |
|---|---|---|---|---|---|---|---|---|---|---|---|---|
| | Dec | Jan | Feb | Mar | Apr | May | Jan | Feb | Mar | Apr | | |
| Bacillariophyta: Coscinodiscophyceae | 0 (0) | 1.5 (2.1) | 6.2 (9.4) | 12.1 (30) | 9.6 (13.2) | 1.9 (4.9) | 1.6 (1.9) | 0.6 (0.7) | 0.2 (0.5) | 1.7 (3.9) | 561 | 18.1 |
| Crustacea: Copepoda adult | 37.5 (14.8) | 4 (5.7) | 5.5 (4.8) | 3.2 (3.1) | 6.1 (13.8) | 1.8 (1.9) | 2.4 (3.8) | 0.9 (1.5) | 0.7 (1) | 3.4 (5.2) | 444 | 14.3 |
| Bacillariophyta: Bacillariophyceae | 0.5 (0.7) | 14.5 (7.8) | 7.8 (15.6) | 1.7 (5.2) | 3.7 (6) | 2.9 (5.9) | 1.6 (1.7) | 1.6 (2.5) | 1.3 (2) | 5.4 (8.1) | 395 | 12.8 |
| Chordata: Fish (eggs & larvae) | 1 (0) | 6 (1.4) | 3.1 (3.8) | 3.6 (4.1) | 3.2 (6.2) | 3.9 (10.6) | 0.6 (1.3) | 1.9 (2.5) | 0.7 (1.3) | 2.1 (2.7) | 308 | 9.9 |
| Crustacea: Unidentified | 0 (0) | 0 (0) | 0.9 (3.3) | 0.7 (2.3) | 2.7 (4.1) | 1.8 (2.2) | 1.6 (1.5) | 1.8 (3.2) | 4.3 (3.2) | 5.4 (3.7) | 286 | 9.2 |
| Mollusca: Bivalve larvae | 0 (0) | 2 (2.8) | 2.2 (2.8) | 3.7 (5) | 1.9 (4.3) | 0.8 (1.4) | 1 (1) | 2.5 (2.5) | 0.6 (0.8) | 2.8 (3.3) | 229 | 7.4 |
| Mollusca: Gastropod larvae | 0 (0) | 6.5 (9.2) | 3.4 (4.7) | 1.9 (3.7) | 1.4 (2.3) | 0.7 (1.5) | 0 (0) | 3.2 (3.4) | 0.5 (1.4) | 0.2 (0.4) | 193 | 6.2 |
| Crustacea: Cirripedia larvae | 1 (0) | 0 (0) | 0.8 (1.3) | 0.5 (1.1) | 1.5 (2.4) | 0 (0) | 0.6 (0.5) | 1.3 (1.9) | 0.3 (0.8) | 0.3 (0.7) | 88 | 2.8 |
| Chaetognatha: Sagittoidea | 1 (1.4) | 0 (0) | 0.2 (0.8) | 0.7 (1.3) | 1.3 (2.1) | 0.5 (1.2) | 2.6 (2.4) | 0.4 (0.9) | 0 (0.2) | 0.3 (0.7) | 75 | 2.4 |
| Ciliophora | 0 (0) | 0 (0) | 0 (0) | 0 (0) | 0.6 (1.6) | 0 (0) | 6.4 (14.3) | 0 (0) | 0 (0) | 0 (0) | 46 | 1.5 |
| Annelida: Polychaeta larvae/juvenile | 0 (0) | 0 (0) | 0.1 (0.3) | 0.3 (1) | 0.1 (0.3) | 0.2 (0.8) | 0.4 (0.5) | 0.3 (0.7) | 0.4 (0.8) | 0.3 (0.7) | 31 | 1.0 |
| Crustacea: Brachiura larvae | 0 (0) | 1 (1.4) | 0.5 (0.9) | 0.1 (0.3) | 0.2 (0.7) | 0.2 (0.6) | 0 (0) | 0.3 (0.4) | 0.1 (0.3) | 0.1 (0.3) | 25 | 0.8 |
| Nematoda | 0 (0) | 0 (0) | 0 (0) | 0.1 (0.5) | 0 (0.2) | 0.3 (0.6) | 0.2 (0.4) | 0.4 (0.7) | 0.2 (0.5) | 0 (0) | 19 | 0.6 |
| Crustacea: Amphipoda adult | 1 (1.4) | 0 (0) | 0 (0) | 0 (0) | 0 (0.2) | 0 (0) | 0.2 (0.4) | 0 (0) | 0 (0.2) | 0.3 (1) | 8 | 0.3 |
| Crustacea: Anomura (*Emerita* sp) adult | 0 (0) | 0 (0) | 0 (0) | 0 (0) | 0.2 (0.5) | 0 (0) | 0 (0) | 0 (0) | 0 (0) | 0 (0) | 4 | 0.1 |
| Unidentified | 5.5 (2.1) | 11.5 (6.4) | 4.5 (4.6) | 4.3 (4) | 2.2 (3.4) | 3.2 (4.8) | 4.8 (3.8) | 3 (2.9) | 2.5 (3.1) | 0.6 (0.9) | 386 | 12.5 |
| Average number of prey items | 47.5 (9.3) | 47 (4.5) | 35.2 (2.6) | 32.8 (3.1) | 34.7 (2.6) | 18.3 (1.3) | 24 (1.8) | 18.2 (1.1) | 12 (1.1) | 23.1 (1.9) | 29.3 | |

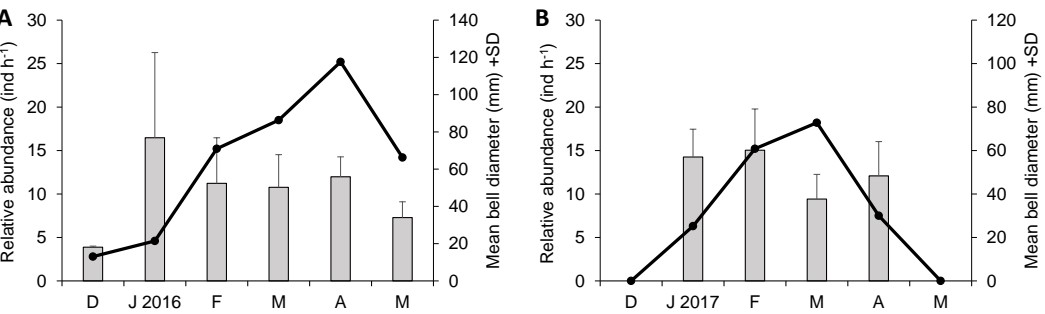

**Figure 2** **Monthly variability of bell diameter and relative abundance of *Stomolophus meleagris* in La Plata Archipelago (Málaga Bay, Colombia) during two medusoid seasons.** Grey bars represent bell diameter and black lines the relative abundance of Stomolophus meleagris. Seasons are: December 2015–May 2016 (A); December 2016–May 2017 (B)

sample size was smaller. Therefore, samples from December 2015 ($N = 2$) January 2015 ($N = 2$) and January 2016 ($N = 5$) were excluded from the analysis. Finally, when temporal differences were confirmed by the PERMANOVA, a Canonical Ordination of Principal Coordinates (CAP, *Anderson & Willis, 2003*) was used as a constrained ordination that best defines groups (months) according to the diet structure.

To assess if the body sizes of medusae and their symbionts are related, length data were fitted to linear, polynomial, logarithmic, exponential and power models. The best fit was chosen according to the proportion of variance explained. Models were fitted by least squares procedures using the algorithm Levenberg–Marquardt to estimate standard errors (SE) of the parameters. Finally, we performed a literature review of the reported symbiotic fauna for *S. meleagris* to compare the diversity of associations found in the study area. The terms "*Stomolophus*" and "*Stomolophus meleagris*", excluding the terms "venom", "protein" and proteomics" from the title or in combination with "association", "symbiosis" or "relationship", were used to search the ISI Web of Science database and Google Scholar. The resulting literature was then manually scanned for descriptions of symbiotic relationships

## RESULTS

### Jellyfish seasonality

Two species of scyphozoan jellyfish were found during this study: *S. meleagris* and *Pelagia noctiluca*. The latter was found only occasionally; one individual during season 1 and six during the season 2. Such a small sample size circumvents any quantitative analysis and therefore only descriptive details will be given. The relative abundance of *S. meleagris* consistently showed a unimodal pattern with peaks during March or April (Fig. 2). Moreover the bell diameter did not show a consistent pattern of growing or decreasing size over time.

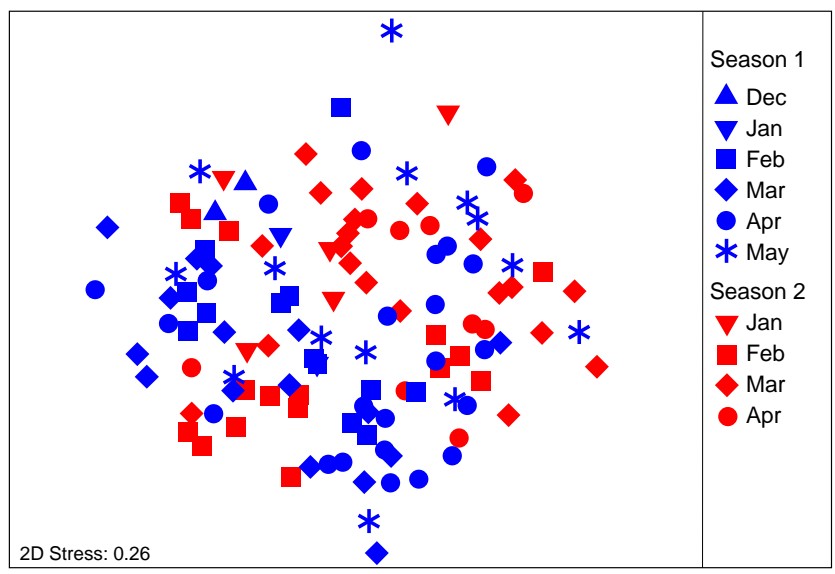

**Figure 3** **nMDS ordination plot on the diet composition of *Stomolophus meleagris*.** nMDS calculated from Bray–Curtis dissimilarity measures with square-root transformed data of abundance per food item during two medusoid seasons (December 2015–May 2016 and Jan 2017–Apr 2017).

## The dynamics of food consumption by *S. meleagris*

The composition and abundance of food items consumed by *S. meleagris* is shown in Table 1. The diet was prevalently comprised of bacillariophyte algae, copepods and fish early stages. Small amounts of benthic items (e.g., juvenile polychaetes, amphipods and adult *Emerita* sp) were found frequently and suggest that *S. meleagris* feeds near the bottom. A few nematods, typically jellyfish parasites, were also found but included as food items because they were not buried in the host's tissue, as observed by *Phillips & Levin (1973)*, and showed obvious signs of digestion. Arguable between-month differences in diet composition were apparent form the nMDS ordination (Fig. 3), particularly when the same month from different seasons were compared (e.g., March in each season). However, as the PERMDISP test was significant for the factor "month" ($F_{9,111} = 4.242$; $P_{(perm)} = 0.006$), a PERMANOVA might yield misleading results. To circumvent this problem, months with small sample sizes ($N \leq 5$) were excluded (PERMDISP test not significant: month: $F_{6,105} = 2.183$; $P_{(perm)} = 0.066$; season: $F_{1,110} = 1–734$; $P_{(perm)} = 0.211$).

The PERMANOVA analysis (Table 2A) showed significant differences in the structure of the diet of *S. meleagris* among months ($P_{(perm)} = 0.001$) but not between seasons ($P_{(perm)} = 0.134$). Pair-wise comparisons (Table 2B) revealed between-month differences, which were best illustrated by the CAP constrained ordination (Fig. 4). Scyphozoan jellyfish are widely considered carnivore predators and, as such, the inclusion of phytoplankton items in the analysis of diet composition seems unwarranted and the described dynamics of food consumption questionable. Therefore, the PERMANOVA analysis was re-run with exclusion of the Coscinodiscophyceae and Bacillariophyceae. This analysis rendered

**Table 2** Results of (A) the PERMANOVA analysis on the differences in the structure of the diet of *Stomolophus meleagris* among months and seasons and (B) pair-wise tests for differences between pairs of months in each season.

| A | Source | df | SS | MS | Pseudo-*F* | P (perm) | Unique perms |
|---|--------|----|----|----|-----------|----------|--------------|
| | Season | 1 | 10,841 | 10,841 | 2.2673 | 0.134 | 917 |
| | Month(season) | 5 | 24,981 | 4,996 | 2.4719 | **0.001** | 998 |
| | Res | 105 | 212,230 | 2,021 | | | |
| | Total | 111 | 249,810 | | | | |

| B | Months | Season | T | P (perm) | perms |
|---|--------|--------|---|----------|-------|
| | February, March | 1 | 1.440 | 0.077 | 999 |
| | February, April | 1 | 1.427 | 0.087 | 999 |
| | February, May | 1 | 1.395 | 0.083 | 998 |
| | March, April | 1 | 1.717 | **0.017** | 999 |
| | March, May | 1 | 1.510 | **0.041** | 999 |
| | April, May | 1 | 0.982 | 0.486 | 997 |
| | February, March | 2 | 2.101 | **0.004** | 999 |
| | February, April | 2 | 1.830 | **0.016** | 998 |
| | March, April | 2 | 1.270 | 0.151 | 999 |

**Notes.**
Significant factors ($\alpha = 0.05$) are highlighted in bold.

remarkably similar results: significant differences in the structure of the diet among months ($P_{(perm)} = 0.002$) but not between seasons ($P_{(perm)} = 0.128$) (Table S1).

## Traits and diversity of the biological associations

The body size of the 121 collected *S. meleagris* ranged between 12.1 and 109.2 mm in bell diameter. The prevalence of symbiotic associations (i.e., the percentage of medusa harbouring symbionts) was high, with 50.4% of the collected specimens. With only three exceptions, a single symbiont per medusa was found, clearly indicating that the distribution of symbionts among host was not random but uniform. The symbiotic community of *S. meleagris* was composed of fishes (95.3%), crustaceans (3.1%) and molluscs (1.6%) (Fig. 5). Generally, associated fish reacted to disturbance by hiding within oral arms or below the host's bell; it could be said that fish "resist efforts to separate them", as stated by *Hargitt (1904)*. However, the prevalence of symbiotic association might have been underestimated, because some may have escaped during the sampling. Regularly only one symbiont per host was found, with only three exceptions, where two fish per medusa where observed. *P. noctiluca* also had symbiotic associations: two out of six medusae harboured individual fish (*Hemicaranx zelotes*).

The body size of the scyphozoan *S. meleagris* showed a significant positive correlation with the body size of its symbiotic community as a whole, the fish assemblage and *H. zelotes* in particular (Fig. 5; Table 3). Power models best fitted the positive body size relationships, and the model including only *H. zelotes* had the highest proportion of variability explained (0.634). This partially reflects the fact that *H. zelotes* was the most common symbiont of *S. meleagris*. Figure 5 shows that the symbionts were generally smaller than the host.

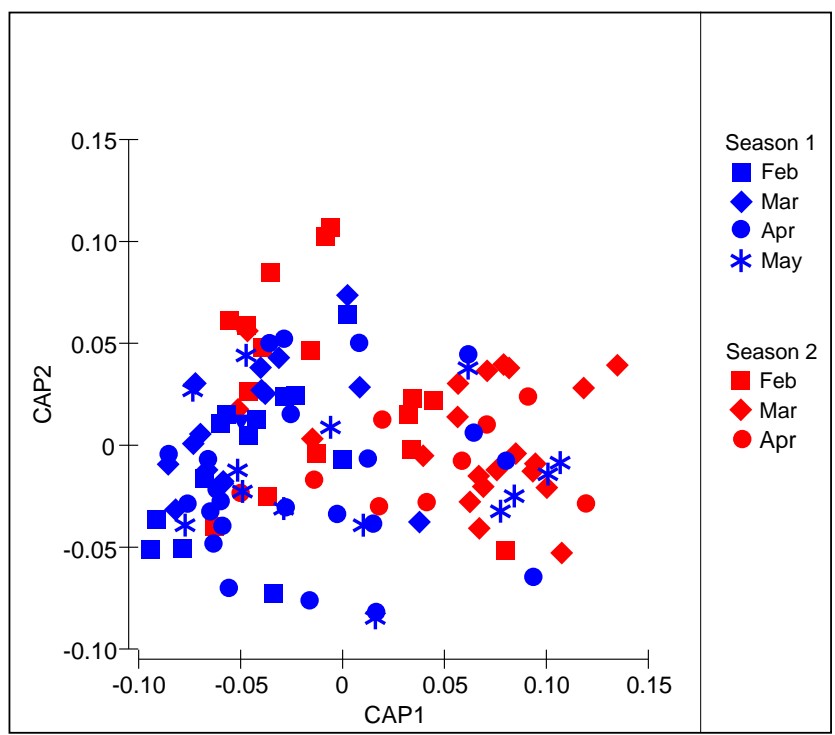

**Figure 4** **Constrained Canonical Analysis of Principal Coordinates of the diet composition of *Stomolophus meleagris*.** CAP analysis based in Bray–Curtis dissimilarity measures with square-root transformed data of abundance per food item during two medusoid seasons (December 2015–May 2016 and Jan 2017–Apr 2017).

However, less frequent symbionts did not follow that pattern; for instance, *Centropomus medius*, *Lutjanus guttatus*, *Oligoplites altus*, *Selene brevoortii* and *Gerres similimus* were larger, and *Hyporhamphus snyderi* almost twice as large as its host.

The richness of the symbiotic fauna reported for *S. meleagris* in the study area was unexpectedly high: 11 species, 10 of them being new reports of symbionts for this species. This richness represents 39.2% of the total diversity of associations found so far (28 symbiotic species reported; Table 4) for this widely-distributed scyphozoan jellyfish.

## DISCUSSION

The occurrence of *S. meleagris* in the study area showed a consistent seasonal pattern that coincides with the seasonal increase in the sea surface temperature from January to May in the study area (*IDEAM, 2004*). The body size did not show an increasing trend through the seasons. This suggest that the observed medusae did not recruit from local benthic polyps, but arrive to the estuarine system as adults from adjacent areas, as discussed by *Kraeuter & Seltzler (1975)* for *S. meleagris* in Georgian and North Carolina waters.

There are only three published studies on feeding ecology of *S. meleagris*. Taken together, these studies highlight that a few taxa form a high percentage of the total gut content. *Larson (1991)* found that in the north-eastern Gulf of Mexico 98% of *S. meleagris* diet

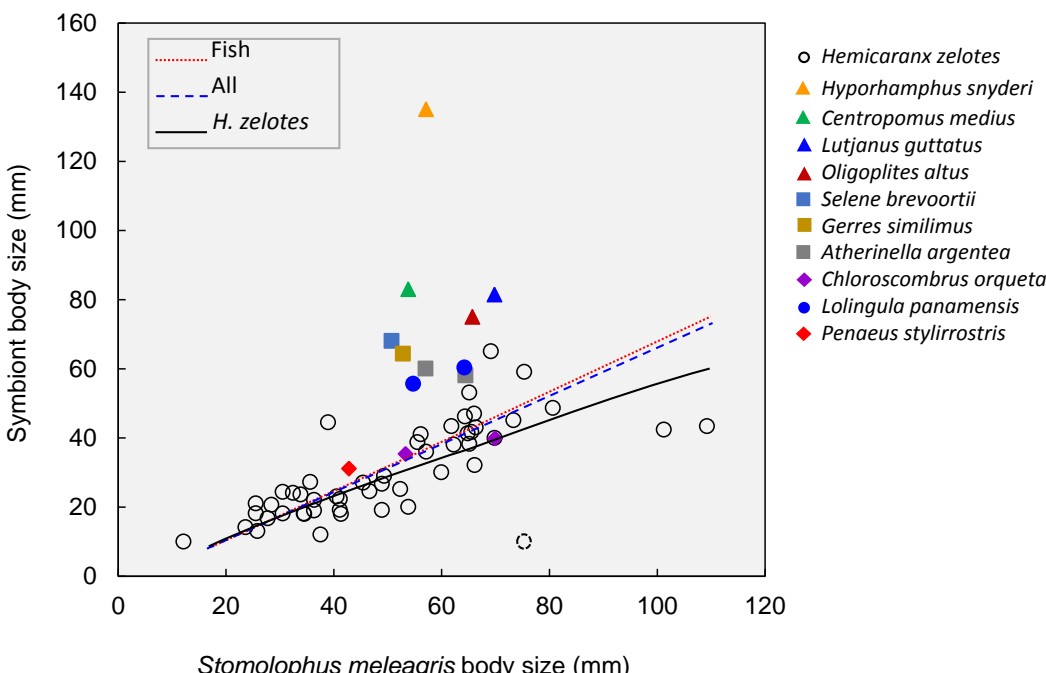

**Figure 5** **Body size relationships between *Stomolophus meleagris* and its symbiotic fauna.** Lines represent the model fits for *Hemicaranx zelotes*, fishes and the whole symbiotic community. Parameter estimations and associated statistics for each model are given in Table 3. The dotted lined circle represents a *H. zelotes,* excluded from the analysis, as its association to the respective medusa could not be confirmed with certainty.

**Table 3** **Results of the model fitting procedure on the relationship of the body size of *Stomolophus meleagris* and its symbionts.**

| Associates | Model | *a* (SE) | *b* (SE) | Pr. Var. | *F*-value | *p* |
|---|---|---|---|---|---|---|
| Fish | $Y = aX^b$ | 1.551 (1.202) | 0.809 (0.189) | 0.291 | 134.763 | <0.001 |
| All | $Y = aX^b$ | 1.775 (1.371) | 0.775 (0.189) | 0.269 | 138.378 | <0.001 |
| *H. zelotes* | $Y = aX^b$ | 1.204 (0.486) | 0.830 (0.098) | 0.634 | 412.441 | <0.001 |

**Notes.**
Pr.Var., Proportion of variability explained by the model.

was composed of bivalve veligers, tintinnids, copepods, gastropod veligers and Oikopleura (Appendicularia). In the Gulf of California, off the coast of Sonora, gut contents were dominated by fish eggs (ca. 83% in a study by *Padilla-Serrato et al., 2013*; and ca. 59% in a study by *Álvarez Tello, López-Martínez & Lluch-Cota, 2016*), followed by mollusc larvae (~26%) and copepods (~11%) (*Álvarez Tello, López-Martínez & Lluch-Cota, 2016*). Although our results (Table 1) also show that a few items comprise high percentages of the diet, the composition and relative importance of those items varied strongly among studies, suggesting strong spatial–temporal differences in food composition. The main difference with previous studies is the consistency of Bacillariophyta among ingested items, surprising for a scyphozoan considered carnivore. However, the fact that phytoplankton items did not

**Table 4** **List of published (a) and new (b) reports on symbionts of the cannonball jellyfish *Stomolophus meleagris*.** An unidentified cestode larva was reported by *Phillips & Levin (1973)*, though it is not included in the list.

| Class: family | Species | Locality and literature source |
| --- | --- | --- |
| (a) | | |
| Actinopterygii: Carangidae | *Chloroscombrus chrysurus* (Linnaeus, 1766) | Mississippi Sound, Mississippi, USA (*Phillips, Burke & Keener, 1969*) |
| | | Wrightsville Beach, North Carolina, USA (*Rountree, 1983*) |
| | | Onslow Bay, North Carolina, USA (*Shanks & Graham, 1988*) |
| | | Texas USA (*Baughman, 1950*) |
| | *Hemicaranx amblyrhynchus* (Cuvier, 1833) | Western Golf of Mexico, USA & Mexico (*Hildebrand, 1954*) |
| | *Hemicaranx zelotes* Gilbert, 1898 | Kino Bay, Sonora, Mexico (*López & Rodríguez, 2008*) |
| | | Málaga Bay, Pacific coast, Colombia (This study) |
| | *Caranx crysos* (Mitchill, 1815) | Barataria Bay, Louisiana, USA (*Gunter, 1935*) |
| | *Caranx hippos* (Linnaeus, 1766) | Wrightsville Beach, North Carolina, USA (*Rountree, 1983*) |
| | *Carangoides bartholomaei* (Cuvier, 1833) | Wrightsville Beach, North Carolina, USA (*Rountree, 1983*) |
| Actinopterygii: Stromateidae | *Peprilus triacanthus* (Peck, 1804) | Wrightsville Beach, North Carolina, USA (*Rountree, 1983*) |
| | | Beaufort, North Carolina, USA (*Smith, 1907*) |
| | | Port Aransas, Texas, USA (*Hoese, Copeland & Miller, 1964*) |
| | | Western Golf of Mexico, USA & Mexico (*Hildebrand, 1954*) |
| | | Gulf of Mexico, USA (*Horn, 1970*) |
| | *Peprilus burti* Fowler, 1944 | Gulf of Mexico, USA (*Horn, 1970*) |
| | *Peprilus paru* (Linnaeus, 1758) | Mississippi Sound, Mississippi, USA (*Phillips, Burke & Keener, 1969*) |
| | | Port Aransas, Texas, USA (*Hoese, Copeland & Miller, 1964*) |
| Actinopterygii: Monacanthidae | *Stephanolepis hispida* (Linnaeus, 1766) | Mississippi Sound, Mississippi, USA (*Phillips, Burke & Keener, 1969*) |
| | | Wrightsville Beach, North Carolina, USA (*Rountree, 1983*) |
| | | Onslow Bay, North Carolina, USA (*Shanks & Graham, 1988*) |
| | *Aluterus schoepfii* (Walbaum, 1792) | Wrightsville Beach, North Carolina, USA (*Rountree, 1983*) |
| | | Woods Hole, Massachusetts, USA (*Hargitt, 1904*) |
| Actinopterygii: Nomeidae | *Nomeus gronovii* (Gmelin, 1789) | Beaufort, North Carolina, USA (*Smith, 1907*) |
| | | Japan & Hong Kong (*Morton, 1989*) |
| Malacostraca: Epialtidae | *Libinia dubia* H. Milne Edwards, 1834 | Beaufort, North Carolina, USA (*Gutsell, 1928*) |
| | | Mississippi Sound, Mississippi, USA (*Phillips, Burke & Keener, 1969*) |
| | | South Carolina, USA (*Corrington, 1927*) |
| | | Wrightsville Beach, North Carolina, USA (*Rountree, 1983*) |
| | | Onslow Bay, North Carolina, USA (*Shanks & Graham, 1988*) |
| | | Fort Pierce, Florida, USA (*Tunberg & Reed, 2004*) |
| | *Libinia* sp | Texas coast, USA (*Whitten, Rosene & Hedgepeth, 1950*) |
| | *Libinia emarginata*, Leach, 1815 | Western Golf of Mexico, USA & Mexico (*Hildebrand, 1954*) |

**Table 4** (*continued*)

| Class: family | Species | Locality and literature source |
|---|---|---|
| Malacostraca: Portunidae | *Charybdis (Charybdis) feriata* (Linnaeus, 1758) | Japan & Hong Kong (*Morton, 1989*) |
| Hexanauplia: Lepadidae | *Conchoderma virgatum* Spengler, 1789 | Guaymas, Mexico (*Álvarez Tello, López-Martínez & Rodríguez-Romero, 2013*) |
| Cestoda: Otobothriidae | *Otobothrium dinoi* (Mendez, 1944) Palm, 2004 | Cananéia, Sao Paulo, Brazil (*Vannucci, 1954*) |
| (b) | | |
| Actinopterygii: Carangidae | *Chloroscombrus orqueta* Jordan & Gilbert, 1883 | Málaga Bay, Pacific coast, Colombia |
| | *Oligoplites altus* (Gunther, 1868) | Málaga Bay, Pacific coast, Colombia |
| | *Selene brevoortii* (Gill, 1863) | Málaga Bay, Pacific coast, Colombia |
| Actinopterygii: Atherinopsidae | *Atherinella argentea* Chernoff, 1986 | Málaga Bay, Pacific coast, Colombia |
| Actinopterygii: Gerreidae | *Gerres simillimus* Regan 1907 | Málaga Bay, Pacific coast, Colombia |
| Actinopterygii: Centropomidae | *Centropomus medius* Günther, 1864 | Málaga Bay, Pacific coast, Colombia |
| Actinopterygii: Hemiramphidae | *Hyporhamphus snyderi*, Meek & Hildebrand, 1923 | Málaga Bay, Pacific coast, Colombia |
| Actinopterygii: Lutjanidae | *Lutjanus guttatus* (Steindachner, 1869) | Málaga Bay, Pacific coast, Colombia |
| Malacostraca: Penaeidae | *Penaeus stylirostris* Stimpson, 1871 | Málaga Bay, Pacific coast, Colombia |
| Cephalopoda: Loliginidae | *Lolliguncula (Lolliguncula) panamensis* Berry, 1911 | Málaga Bay, Pacific coast, Colombia |

influence the temporal patterns in the structure of gut contents suggest a level of structural redundancy (i.e., many items are interchangeable in the way they define changes in composition through time, sensu *Clarke & Warwick, 1998*). However, finding an ingested item does not mean that it is digested, which is one general limitation of studying feeding patterns of jellyfish by gut contents (*Pitt, Connolly & Meziane, 2009*). As Bacillariophyceae (*Nitzschia*) and Coscinodiscophyceae (*Coscinodiscus, Rhizosolenia*) are among the most abundant phytoplankton components in Málaga Bay (*Prahl, Cantera & Contreras, 1990*), it seems reasonable to apply Ockham's principle and assume that the presence of algae in gut contents only reflect their abundance in the water column. Interestingly, *Larson (1991)* also states that ''*Coscinodiscus* sp. was abundant in gut contents'' but it was not listed as prey taxa. It is worth noting that the assumption that jellyfish feed on mesozooplankton and ichthyoplankton is probably related with the essentially arbitrary use of 60–100 µm sieves to concentrate the samples for gut content analysis. In fact, when alternative methods like grazing experiments, microvideographic techniques, stable isotopes and fatty acid tracers are used it becomes apparent that jellyfish can also feed on microzooplankton (*Sullivan & Gifford, 2004*; *Colin et al., 2005*), demersal zooplankton (*Pitt et al., 2008*) and resuspended organic matter (*Javidpour et al., 2016*). Therefore, our results should be regarded as a partial depiction of the diet composition of *S. meleagris*.

As body size of *S. meleagris* did not show a consistent pattern of growth over time (Fig. 2), the intra-season variability observed in the structure of the diet could not be attributed to ontogenetic changes in food habits. In fact, the diet composition of the smaller and larger medusae observed in December 2015 and January 2016, respectively, (Fig. 2), was very similar (Fig. 3). Therefore, the observed intra-season variability might be related with changes in the availability of prey in the water column, but information to evaluate this hypothesis is lacking.

The lack of significant differences in the diet structure of *S. meleagris* between seasons was surprising because the first medusoid season coincided with the major El Niño-La Niña cycle 2015–2016, which heavily modified temperature and rainfall patterns in the study area (*Riascos, Cantera & Blanco-Libreros, 2018*). For example, it is known that the strong modification of freshwater nutrient subsidies through precipitation in Málaga Bay drives changes in the population dynamics and reproductive cycle of benthic estuarine bivalves (*Riascos, 2006*; *Riascos, Heilmayer & Laudien, 2008*). Hence, it would be reasonable to expect shifts in the abundance and composition of the zooplankton community associated to El Niño-La Niña 2015–2016 during the first season, which would be then reflected in significant changes in the diet structure of *S. meleagris* between seasons. It is difficult to speculate on reasons for this result, but perhaps El Niño-La Niña modified the composition of the zooplankton community at lower taxonomic levels (e.g., species, families), which we were not able to detect owed to our categorization of prey items at higher taxonomic levels (Table 1).

Marshes, mangrove forests and seagrass meadows have long been recognized as nursery grounds, mainly because they have extremely high primary and secondary productivity and support a great abundance and diversity of early life stages of fish and invertebrates (*Beck et al., 2001*). Recently, *Doyle et al. (2014)* analysed the role of jellyfish as "service providers" in pelagic habitats and described jellyfish as habitat and nurseries, because they are: (i) larger than most planktonic organisms, (ii) slower swimmers than most nektonic animals and (iii) their diverse morphology provide three-dimensional space for refuge or shelter. Clearly those facts alone do not meet the premise that "a habitat is a nursery for juveniles of a particular species if its contribution per unit area to the production of individuals that recruit to adult populations is greater, on average, than production from other habitats in which juveniles occur" (*Beck et al., 2001*). Strictly speaking, this is a hypothesis remaining to be tested, though some of our results suggest that *S. meleagris* occurring in estuarine systems provide a valuable resource that may significantly increase the survivorship and recruitment of juvenile fishes or invertebrates. First, the great dominance of *H. zelotes* among a diversity of other symbionts suggest a higher suitability to its host. This is in line with the fact that fish of the family Carangidae are the most commonly reported symbiont of *S. meleagris* (Table 4). Secondly, the high prevalence of an association and a uniform distribution of the symbiont within the host population as those observed for *H. zelotes* and *S. meleagris* strongly suggest intraspecific and interspecific interactions and territorial behaviour (*Connell, 1963*; *Britayev et al., 2007*; *Riascos et al., 2011*). And thirdly, positive symbiont-host size relationships, as those observed when *H. zelotes*, the fish assemblage and the whole symbiotic assemblage are analysed, suggest either parallel growth of the host and the symbiont (*Britayev & Fahrutdinov, 1994*) or size-segregation behaviour by the symbiont (*Adams, Edwards & Emberton, 1985*; *Hobbs & Munday, 2004*).

Ecological theory predicts that competition and the "economic defendability" of a resource (*sensu Brown, 1964*) facilitate or hinder the evolution of territoriality; resources are monopolized whenever the benefits exceed the costs of defence. Individuals of a territorial species that fail to obtain a limited resource often make no contribution to future generations (*Begon, Harper & Townsend, 2006*). For jellyfish-fish associations, there

is correlational data suggesting that the shelter and/or food provided by jellyfish increase the survival of juvenile fish to adulthood (*Lynam & Brierley, 2007*). In this context, if the seasonal occurrence of *S. meleagris* does represent a defendable resource, and the influence on the survival of its symbiotic fauna could be experimentally demonstrated, this species may be considered a floating nursery.

According to *Castro, Santiago & Santana-Ortega (2002)*, 333 species of fish belonging to 96 families show aggregative or associative behaviour with floating algae, gelatinous zooplankton, whales, flotsam or man-made fish aggregating devices and 14 of these families associate with jellyfish. Therefore, one may reasonably argue that if the jellyfish-fish association have a measurable effect on fish populations, it can be considered marginal. But how complete is our knowledge of these associations? Regarding *S. meleagris*, Table 4 hints on this question. First of all, it shows that only a few areas of the species distribution range have been studied, particularly the western coast of United States and the Gulf of Mexico. Second, and more importantly, the fact that the findings of our short-term study performed in a small tropical estuarine system represent ca. 40% of the known diversity of the symbiotic fauna of *S. meleagris* strongly suggest that diversity of symbionts increase toward tropical areas and that it is heavily underestimated. Indeed, the seven-year monitory of the bycatch in the trawl fishery of *S. meleagris* off Georgia by *Page (2015)*, rendered 38 species of finfish and three species of invertebrates. Of course, these cannot per se be assumed to be symbionts of *S. meleagris*. But the fact that three species known to be common associates (*Peprilus paru*, *P. triacanthus* and *Chloroscombrus chrysurus*; *Phillips, Burke & Keener, 1969*; *Rountree, 1983*) comprised 63% of the bycatch strongly suggest that some of the other species may actually be unrecognised symbionts. To conclude, a more precise account of the diversity of symbiotic fish-jellyfish associations and an evaluation of their ecological significance may provide a more balanced view of the relationship between fish and jellyfish in marine ecosystems.

## ACKNOWLEDGEMENTS

Thanks to the "Comunidad Negra de La Plata Bahía Málaga" for their hospitality and for allowing access to their beautiful territory. Thanks to Lazarus JF and Cantera JR, for their help in identifying crustaceans and molluscs, respectively. This is the publication number 001 of the Instituto de Ciencias del Mar y Limnología (INCIMAR), Universidad del Valle.

### Funding

Funding for this research was provided by Universidad del Valle. This research was supported by a scholarship to José M. Riascos by the Departamento Administrativo de Ciencia, Tecnología e Innovación (Colciencias) - Programa "Tiempo de Volver". The funders had no role in study design, data collection and analysis, decision to publish, or preparation of the manuscript.

## Grant Disclosures

The following grant information was disclosed by the authors:

Universidad del Valle.

Departamento Administrativo de Ciencia, Tecnología e Innovación (Colciencias).

## Competing Interests

The authors declare there are no competing interests.

## Author Contributions

- José M. Riascos conceived and designed the experiments, performed the experiments, analyzed the data, prepared figures and/or tables, authored or reviewed drafts of the paper, approved the final draft.
- Willington Aguirre conceived and designed the experiments, performed the experiments, authored or reviewed drafts of the paper, approved the final draft, contributed traditional knowledge about local populations of jellyfish in the study area.
- Charlotte Hopfe performed the experiments, analyzed the data, prepared figures and/or tables, authored or reviewed drafts of the paper, approved the final draft.
- Diego Morales and Ángela Navarrete performed the experiments, contributed reagents/materials/analysis tools, prepared figures and/or tables, authored or reviewed drafts of the paper, approved the final draft.
- José Tavera analyzed the data, authored or reviewed drafts of the paper, approved the final draft, identified fish species.

## Animal Ethics

The following information was supplied relating to ethical approvals (i.e., approving body and any reference numbers):

During the course of the field work, fish and a cephalopod associated with jellyfish were incidentally caught. Associated animals were deposited in the scientific collection of marine biology at Universidad del Valle.

## Field Study Permissions

The following information was supplied relating to field study approvals (i.e., approving body and any reference numbers):

Samplings were approved by the Autoridad Nacional de Licencias Ambientales (ANLA; permit number 1070_28-08-2015).

## Data Availability

The raw data are provided in the Supplemental Files.

## Supplemental Information

Supplemental information for this article can be found online at http://dx.doi.org/10.7717/peerj.5057#supplemental-information.

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
