# Peer review of "Floating nurseries? Scyphozoan jellyfish, their food and their rich symbiotic fauna in a tropical estuary"

_PeerJ, doi:10.7717/peerj.5057_

## Round 0.1 · original submission · Major Revisions

Dear Authors,

I have considered the reviewers comments and whilst some see merit in the study and have praise for the manuscript there has been some strong suggestions that more work is required to get this into a publishable form. Therefore, my recommendation is for major revisions.

Please pay particular attention to the reviewers concerns and suggestions regarding the interpretation of figures 3&4 and I look forward to receiving a revised version of the manuscript.

Kind regards,

Alex

Reviewer 1 ·

Basic reporting

Structure problems:
_ The Zoological nomenclature was not used appropriated! For example species usually is written in italic words! In the first moment that a species is informed, it is important to inform the investigator that described the species’ diagnoses;
_ Legends is not self-explanatory;
_ There is disorder in the manuscript topics, for example, line 114 to 116 is Result don’t Materials and Methods; line 211 is Discussion doesn’t Result; and all Discussion the authors mentioned again tables and figures that is usual in the topic Results!
_ Figures 3 and 4: it is very difficult to understand some pattern!
_ Table 3: There are several references, so it is usual to use in the Discussion don’t in the Results.

Experimental design

"Original primary research within Aims and Scope of the journal" - Yes;

Methods were not described with sufficient information. For example:
_ Materials and Methods
Lines 125 until 150 - need some references. The authors created all methodology!

Validity of the findings

Lines 131 until 136 – To inform that “were sent to taxonomic experts for identification” is not enough! The authors need to inform each expert and the bibliography used. If the authors were auxiliated by experts, why are taxa no identity such as the classification only “crustacean” or only “fish”. At some point is not possible to identify in others is possible to identify Brachyura larvae. In studies with “the gut contents” is normal to show figures of each prey item such as Gonçalves et al. (2016 – Symbiosis, vol. 69 (3): 193-198). The prey identifies here was not photographed, why?

Additional comments

Dear authors, I regret to inform you that I rejected your manuscript. I feel that the manuscript has merit, however, I identified several problems. However, I would like to offer and encourage the authors to see my comments and questions to a resubmission.

Reviewer 2 ·

Basic reporting

All my comments are summarized in one section below.

Experimental design

see below.

Validity of the findings

see below.

Additional comments

This manuscript is fairly interesting and well written, but I have major two issues with the write-up.

First, the diet study analyses counts of gut contents, including *numerical abundance* of diatoms and phytoplankton. This does not seem justifiable, and analyzing "prey items" by quantifying counts of phytoplankton and fish equally is not ecologically nor statistically valid, so the ANOVA results are suspect.

Second, the paper emphasizes "symbiosis" including in the title, subtitle, yet the data for the associations are a relatively minor part of the manuscript. For most biologists, the phrase "Scyphozoan symbionts" would suggest the endosymbiosis of Cassiopea or Linuche. Although its use is technically correct, I would prefer that the term "Symbiosis" be largely replaced by "association" in the title and body of manuscript.
On that subject, I would like to see the results for new associations *minus* H. zelotes. Using (Fish) fish (Fish + Other) does not allow examination of the patterns for (Other). In the plot of Body Size vs Symbiont community (Figure 4) the pattern is literally obscured by the H. zelotes points. These should be moved to a second panel, and the points made partially transparent so the overlap can be seen. In the table, the new results are buried with literature review. I would like to see those in a separate subsection.

In the plots for Figures 2 and 3, the selection of point shapes and colors is essentially random. There should be some reason to the assignment of these properties, and that assignment should be consistent between the two figures. For example, Month could be coded by shape (using the same shape between Season 1 and 2) and the season could be coded by color (all the same within each season). Or Month could be by color and season by shape. This would allow the reader to look for patterns and correspondences in what currently looks like a random scattergram.

Other minor comments:
13: what makes this a "basal trophic pathway"? Most medusae are high level predators, and not operating near the "base" of the food chain.

23: State of their symbionts: To me this means algal endosymbiosis, since it has not yet been defined and qualified. Please change to "associated" fauna.

26: algae are not a significant contributor to nutrition, so abundance is meaningless. How were diatom chains even enumerated?

29: Associated *with* [not "to"] -- but yes, associated, is the preferred phrase here.

30: Thereby -> Therefore

34: occurrence [fix spelling]

37: consistent abundance of algae: not to be emphasized.

94: cannonball [spelling]

114: permit number instead of permission number?

115: Medusoid occurrence numbers should be in results, not methods

144: Abundance of each taxon: again, not meaningful

177: Two species of scyphozoan jellyfish [more than two jellyfish were found]

199: between month differences [not months]

265: remove "on average" - makes the phrasing confusing and not needed.

275: Were those years really "El Niño/La Niña?" — which one was it, since they alternate.

·

Basic reporting

The manuscript is written clearly and generally with unambiguous text (a few minor changes suggested below for clarity). The tables and figures are appropriate and the raw data is supplied in a suitable format.

Experimental design

I found this to be interesting research and commend the authors on their data collection, I know first-hand how tricky collecting this information is.

Validity of the findings

This work fills an area of research where there is a paucity of information on and is often overlooked. The conclusions are clear and intelligible, as well as supporting the authors’ research question and are supported by the results.

Additional comments

Review of: Floating nurseries? Scyphozoan jellyfish, their food and their rich symbiotic fauna in a tropical estuary
Riascos et al.
I found this to be interesting research and commend the authors on their data collection, I know first-hand how tricky collecting this information is. The manuscript is written clearly and generally with unambiguous text (a few minor changes suggested below for clarity). The tables and figures are appropriate and the raw data is supplied in a suitable format.
This work fills an area of research where there is a paucity of information on and is often overlooked. The conclusions are clear and intelligible, as well as supporting the authors’ research question and are supported by the results.

Minor suggestions:
Line 104: Sampling: A map with the study area and its location in the region would be a useful addition.
Line 107: Change “one of the rainiest places in the western hemisphere” to “an area with one of the highest levels of precipitation in the western hemisphere”
Line 110: Is the tidal range always 4.1m? If this is an average please add ‘mean’ to the text.
Line 181: Change “body” to “bell diameter” for clarity as this is what you have measured, body is too ambiguous.
Line 210: change “below the bell´s host” to “below the host´s bell”.

---

## Round 0.2 · Minor Revisions

Dear Jose,

I've sent this back with minor revisions as a reviewer has suggested some very minor corrections however please consider the manuscript accepted. Thank you for submitting your paper to PeerJ.

Reviewer 1 ·

Basic reporting

.

Experimental design

.

Validity of the findings

.

Additional comments

Dear Authors,
I my opinion, now the manuscript has merit. I just want to suggest two corrections:
1. Please, pattern species in italic words (see line 56);
2. I suggest new nomenclatures in the table 1, because when the authors use “Arthropoda: Crustacea” include all others crustaceans, so my suggestions are:
“Arthropoda: Copepoda” to “Crustacea: Copepoda adult”
“Arthropoda: Crustacea” to “Crustacea: Unidentified”
“Arthropoda: Cirripedia larvae” to “Crustacea: Cirripedia larvae”
“Arthropoda: Brachiuran larvae” to "Crustacea: Brachyura larvae”
“Arthropoda: Amphipoda” to “Crustacea: Amphipoda adult”
“Arthropoda: Decapoda” to “Crustacea: Anomura (Emerita sp) adult”

---

## Round 0.3 · accepted · Accept

Dear Authors

Thank you for adding those extra few amendments and once again for submitting to PeerJ.

#